# Random forest model for feature-based Alzheimer's disease conversion prediction from early mild cognitive impairment subjects

**Matthew Velazquez**📷*, **Yugyung Lee, for the Alzheimer's Disease Neuroimaging Initiative**¶

Department of Computer Science, University of Missouri - Kansas City, Kansas City, MO, United States of America

¶ Membership of the Alzheimer's Disease Neuroimaging Initiative is listed in the Acknowledgments section.
* mv3md@mail.umkc.edu

**Data Availability Statement:** The data underlying the results presented in the study are available from the Alzheimer's Disease Neuroimaging Institute. http://adni.loni.usc.edu/. Other authors

## Abstract

Alzheimer's Disease (AD) conversion prediction from the mild cognitive impairment (MCI) stage has been a difficult challenge. This study focuses on providing an individualized MCI to AD conversion prediction using a balanced random forest model that leverages clinical data. In order to do this, 383 Early Mild Cognitive Impairment (EMCI) patients were gathered from the Alzheimer's Disease Neuroimaging Initiative (ADNI). Of these patients, 49 would eventually convert to AD (EMCI_C), whereas the remaining 334 did not convert (EMCI_NC). All of these patients were split randomly into training and testing data sets with 95 patients reserved for testing. Nine clinical features were selected, comprised of a mix of demographic, brain volume, and cognitive testing variables. Oversampling was then performed in order to balance the initially imbalanced classes prior to training the model with 1000 estimators. Our results showed that a random forest model was effective (93.6% accuracy) at predicting the conversion of EMCI patients to AD based on these clinical features. Additionally, we focus on explainability by assessing the importance of each clinical feature. Our model could impact the clinical environment as a tool to predict the conversion to AD from a prodromal stage or to identify ideal candidates for clinical trials.

## Introduction

Alzheimer's Disease (AD) is a progressive, degenerative brain disorder that leads to nerve cell death and tissue loss in the brain. Currently, there are no treatment plans that prevent the progression of AD, and this has led to increased emphasis on being able to predict AD at an earlier stage. Mild Cognitive Impairment (MCI) is an intermediary stage between being cognitively normal and having AD where 32% of MCI patients will go on to develop Alzheimer's Disease [1]. This makes the MCI stage an ideal target for early prediction as studies point to early diagnosis as being key to potentially delaying the overall progression of AD [1]. Early detection at the MCI stage can assist in clinical trial enrollment and provide more specific treatment plans when more effective ones do become available. Our focus in this study was to target the earliest

will have access to this data in the exact same manner as the authors.

**Funding:** The author(s) received no specific funding for this work.

**Competing interests:** The authors have declared that no competing interests exist.

subset of MCI patients (EMCI), as that subset is the furthest from an AD diagnosis and would thus provide a more beneficial prediction. As a result, it is of high importance to accurately determine which EMCI patients will develop AD.

For this reason, an accurate, ensemble learning model that can aid in clinical decision making is necessary to help ascertain the patient's prognosis. Random Forest A random forest algorithm is a supervised learning algorithm that randomly creates and merges multiple decision trees and that has been proficient with classification problems [2]. In our work, this Random Forest a random forest model is used to determine which patients will convert to AD (our EMCI_C class) and those that will not convert (EMCI_NC) against an imbalanced data set. As well as determining how to best balance the data, assessing which clinical features are most relevant for conversion prediction is fundamental to our problem. Through Random Forest our random forest model we are able to see which of our clinical features has the most significant impact at both the model and the individual prediction levels. This allows us to interpret the individual results better to provide more clinical significance. In this study, we sought to (1) identify significant features from clinical data; (2) build a random forest classification model from an imbalanced data set of those features; (3) determine the prediction accuracy of our model.

Also, we observed the associations between individual predictors and their importance to the problem. By attempting different feature groupings, we were able to distinguish the most crucial feature types. As a result of this approach, our work provides a clinical decision-making tool that can predict MCI-to-AD conversion with high accuracy and interpret the results meaningfully. We envision that this work will provide an accurate tool for predicting conversion probability from MCI to AD and further understand the impact of neuropsychological, biomarker, and demographic features.

## Related work

A review on the use of random forest models in classifying Alzheimer's Disease was provided by Sarica et al. [3]. Their review consisted of 12 studies that were primarily focused on the classification of Alzheimer's Disease stages from MRI images. The accuracy across these studies ranged from 53% to 96%, depending on whether they were performing multiclass classification or not. These studies were also focused on the direct stage classification of AD vs. Normal Controls vs. MCI, rather than the prediction of AD from an earlier stage.

Another review by Weiner et al. [4] summarized 49 ADNI papers. These papers did target the prediction of AD but were also focused mainly on MRI data. These were occasionally supplemented by clinical data or other imaging data (PET), with most studies using a support vector machine (SVM) model. A few of the studies [5–7] did use the random forest algorithm as well and will be compared, alongside the SVM implementations, against our model's performance.

Huang et al. [8] proposed a predictive nomogram that combined image features, clinical factors, and AB concentration to predict the conversion of MCI to AD. They also explored the associations between the different selected features and reported on their significance. Their goal was to examine the associations at both a macro and micro level to better understand the underlying patterns.

Moore et al. [9] proposed using a pairwise selection from time-series data to predict AD conversion. The authors analyzed the relationships between data point pairs at different times using a random forest algorithm. They leveraged a mix of demographic and genetic data and achieved a classification accuracy of 73% as a result.

Lebedev et al. [6] used a combination of structural MRI scans along with a few clinical features from the ADNI data set to achieve an MCI-to-AD conversion accuracy of 81.3%. Their work also saw a sharp increase in accuracy by using a Random Forest algorithm rather than a Support Vector Machine. One advantage in their study is that they validated the model extensively outside of the ADNI data set and found no substantial drop in accuracy, suggesting a good foundation for clinical implementation.

Rana et al. [10] created a model deemed MudNet, which combined both clinical data and MRI imaging for MCI-to-AD conversion prediction. They used many neuropsychological assessment scores alongside T1-weighted structural MRIs to achieve a conversion accuracy of 69.8%. Their work also provided a time-to-AD conversion classification which differentiated between high-risk (AD conversion within two years), and low-risk people (AD conversion greater than two years) at a 66.9% accuracy.

Thushara et al. [11] used a random forest algorithm for multi-class Alzheimer's classification. Their work sought to distinguish between AD, MCI, cMCI (Converted MCI), and normal controls using largely biomarker features. They achieved a multi-class classification accuracy of 69.33% with an MCI-to-AD conversion prediction (cMCI class) accuracy of 47.19%.

## Methods

### Alzheimer's disease neuroimaging initiative data

All data used for this paper were obtained from the Alzheimer's Disease Neuroimaging Initiative (ADNI) database and included patients from their ADNI-1, ADNI-2, and ADNI-GO studies [12]. "The ADNI was launched in 2003 as a public-private partnership with the primary goal of testing whether serial magnetic resonance imaging (MRI), positron emission tomography (PET), other biological markers, and clinical and neuropsychological assessment can be combined to measure the progression of mild cognitive impairment (MCI) and early Alzheimer's disease (AD)" [12]. Early Mild Cognitive Impairment (EMCI) patients were eligible for our study as long as they had follow-up appointments for greater than a year. The EMCI subset consists of patients that are 5-7 years before a possible AD diagnosis and are identified by the results of the Wechsler Memory Scale Logical Memory II test. These EMCI patients were then subdivided into two groups based on whether they would eventually be diagnosed with Alzheimer's Disease or not. We chose to represent these groups as EMCI_C, for our AD conversion group, and EMCI_NC for our stable group. From the ADNI variables, the Clinical Dementia Rating was used to make this determination based on the value of their last visit's diagnosis. The remaining 1806 EMCI visits were then used as a starting point for training prior to augmentation. Of these, 198 belonged to the EMCI_C class while 1608 visits were from EMCI_NC subjects. Overall, our study consisted of 383 EMCI patients (shown in Fig 1), with 49 belonging to the EMCI_C group and the remaining 334 within the EMCI_NC group. These patients were then randomly split such that 75% (288 patients) of our selected patients were used to train the random forest model, with the remaining 25% (95 patients) used for validation testing (shown in Table 1).

### Clinical features selection

The clinical features that were used to train our random forest model included a mix of genetic biomarkers (APOE4), physical biomarkers (hippocampal and ventricular volume), four neuropsychological scale scores (ADAS13, ADAS11, FAQ, MMSE), and the patient's demographic information (age, race). Many different variations of ADNI features were tested for model inclusion; however, these nine features were found to provide the best overall fit. Additionally, related studies have used similar features and found the mix of biomarker and

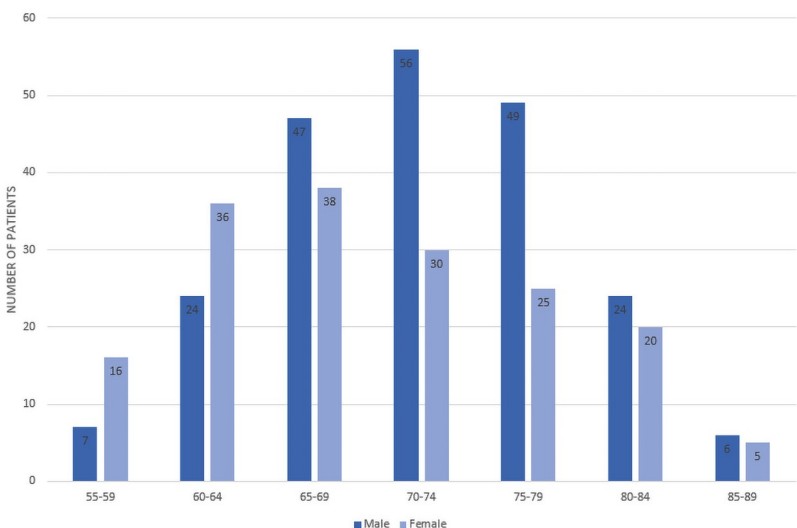

**Fig 1. Participants' age and gender distribution.**

neuropsychological scale scores to be an ideal selection for AD prediction [13]. The ADNI features that we have used per training group can be seen in Table 2.

## Random forest classification model

Random forests are an ensemble learning method for classification, regression, and other tasks that operate by constructing a multitude of decision trees at training time and outputting the class that is either the mode of the classes, in regards to a classifier, or the mean prediction of the individual trees for a regression model. Since random forests consist of a collection of decision trees that are trained with different data subsets and then averaged, this allows them to be tolerant of the problem of overfitting.

For our work (as seen in Fig 2), the random forest classifier has two potential classes eligible for its output, EMCI_C (patients that converted to AD) and EMCI_NC (patients that did not convert to AD). These classes are voted on from each individual tree, which is then aggregated to provide an overall probability of AD conversion. Fig 3 shows an example of an individual tree. Random Forest classifiers also allow for individual input variable importance to be evaluated. As part of our work, we built an ad hoc prediction script that evaluates this variable importance at both the model and individual prediction levels. Initially, while training the model, this evaluation helped us determine which variables were most relevant for model inclusion. After the model has been trained, this variable importance ranking then helps to interpret the individual prediction results Table 3 shows the rankings of 6 features, 9 features

**Table 1. EMCI data set for machine learning.**

|  | EMCI_C | EMCI_NC |
|---|---|---|
| Subject# | 49 | 334 |
| Visit# | 198 | 1608 |
| Record# after Oversampling | 1608 | 1608 |
| Training Data | 1206 | 1206 |
| Testing Data | 402 | 402 |

**Table 2. ADNI clinical features and EMCI patient characteristics used for random forest model training.**

| ADNI Feature | | EMCI_C | | EMCI_NC | | 6FT | 9FT | 13FT |
|---|---|---|---|---|---|---|---|---|
| | Subject# | 49 | | 334 | | | | |
| | Time to AD Conversion | 5.02 years | | – | | | | |
| | Time in Study | 12 years | | 12 years | | | | |
| | Description | Mean | SD | Mean | SD | | | |
| DX | Diagnosis | – | – | – | – | – | – | – |
| *Demographic Info* | | | | | | | | |
| PTRACCAT | Patient Race | – | – | – | – | ✓ | ✓ | ✓ |
| AGE | Patient Age | 73.5 | 6.47 | 71.1 | 7.49 | ✓ | ✓ | ✓ |
| *Genetic Biomarkers* | | | | | | | | |
| APOE4 | #E4 alleles of APOE | .9 | .71 | .4 | .46 | – | ✓ | ✓ |
| *Physical Biomarkers* | | | | | | | | |
| Hippocampus | Hippocampal volume | 6875.2 | 947.45 | 7334.1 | 910.20 | – | ✓ | ✓ |
| Ventricles | Ventricular volume | 39282.7 | 21031.66 | 34504.6 | 21394.49 | – | ✓ | ✓ |
| *Neuropsychological* | | | | | | | | |
| ADAS13 | 13-item AD Assessment Scale | 15.8 | 6.02 | 13.3 | 5.41 | ✓ | ✓ | ✓ |
| ADAS11 | 11-item AD Assessment Scale | 9.7 | 4.12 | 8.5 | 3.29 | ✓ | ✓ | ✓ |
| FAQ | Functional Activities Questionnaire | 4.1 | 4.38 | 1.82 | 2.50 | ✓ | ✓ | ✓ |
| MMSE | Mini-Mental State Examination | 28.1 | 1.58 | 28.3 | 1.71 | ✓ | ✓ | ✓ |
| RAVLT_immed | #words memorized over 5 trials | 34.5 | 8.39 | 40.3 | 11.40 | – | – | ✓ |
| RAVLT_learn | #words learned between trials 1-5 | 4.7 | 2.47 | 5.3 | 2.42 | – | – | ✓ |
| RAVLT_forg | #words forgotten between trials 5-6 | 5.1 | 2.54 | 4.1 | 2.64 | – | – | ✓ |
| RAVLT_perc_forg | %words forgotten between trials 5-6 | 60.7 | 29.13 | 44.0 | 29.36 | – | – | ✓ |

*EMCI_C* the converter group, *EMCI_NC* the stable group, *FT* Feature Training

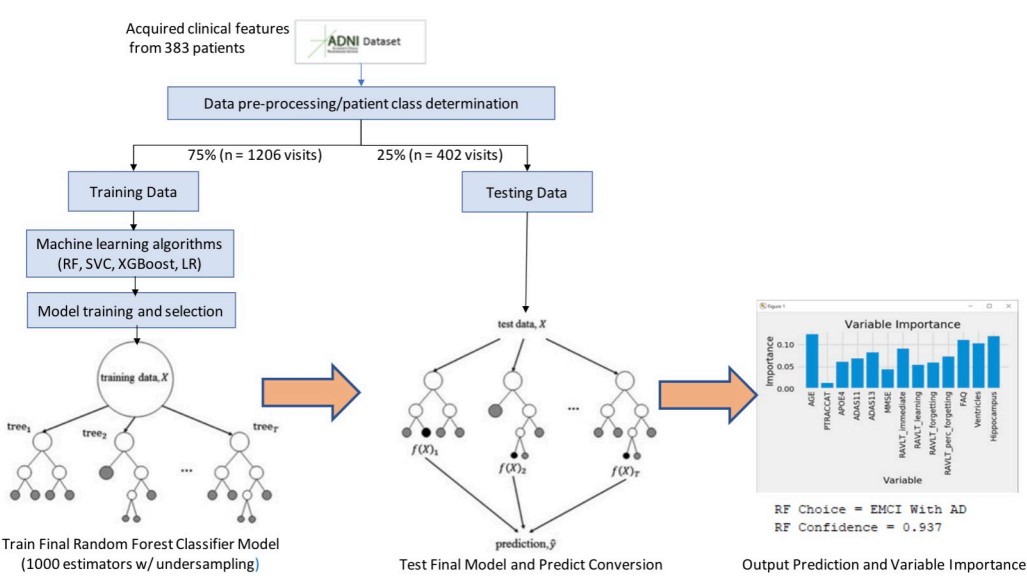

**Fig 2. Model workflow.**

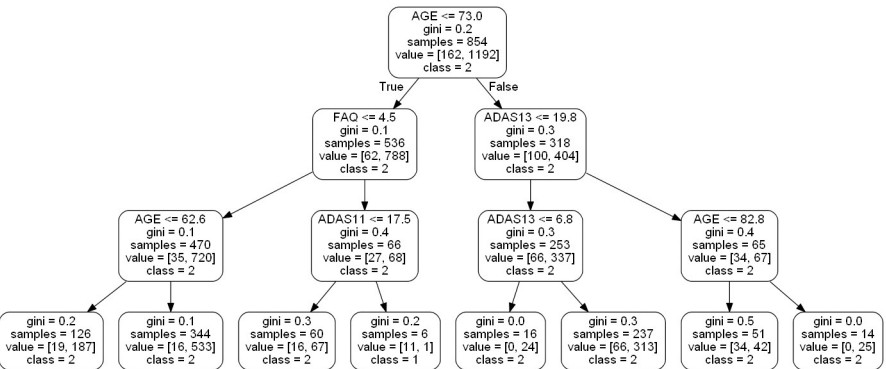

**Fig 3. Example of a small random forest tree within our model.**

and 13 features as well as the fractional ranks that are the average of the ordinal ranks for these three feature groups.

## Results

### Demographic and clinical characteristics

As can be seen across Fig 1 and Table 2, 383 EMCI patients were gathered from the ADNI database, of which 49 would convert to AD (EMCI_C), and 334 would not convert (EMCI_NC). The patients' average age was 71.4, and 55.6% of the patients were men. There was a significant difference in age between the two groups (P < .05) according to the t-test, however this feature's difference was not statistically significant between classes when measuring feature importance. Additionally, a subsequent model was trained on a reduced data set that eliminated Age outliers and accuracy was only reduced by.3%.

Also shown in Table 2 are the genetic and physical biomarkers. The APOE4 and hippocampal volume differences were statistically significant between the EMCI_C and EMCI_NC groups, whereas the ventricular volume was not. For the neuropsychological scale scores, the ADAS13 and the FAQ features were significantly different (P < .05). The ADAS11 and the

**Table 3. Comparison of feature importance ranking by feature groups.**

| Feature | 6FT Ranks | 9FT Ranks | 13FT Ranks | Fractional Ranks |
|---|---|---|---|---|
| AGE | **1** | **1** | **1** | **1** |
| FAQ | 2 | 4 | 3 | 3 |
| ADAS13 | 3 | 5 | 6 | 4.6 |
| ADAS11 | 4 | 6 | 8 | 6 |
| MMSE | 5 | 8 | 12 | 8.3 |
| PTRACCAT | 6 | 9 | 13 | 9.3 |
| Hippocampus | - | 2 | 2 | 2 |
| Ventricles | - | 3 | 4 | 3.5 |
| APOE4 | - | 7 | 9 | 8 |
| RAVLT_immed | - | - | 5 | 5 |
| RAVLT_perc_forg | - | - | 7 | 7 |
| RAVLT_forg | - | - | 10 | 10 |
| RAVLT_learn | - | - | 11 | 11 |

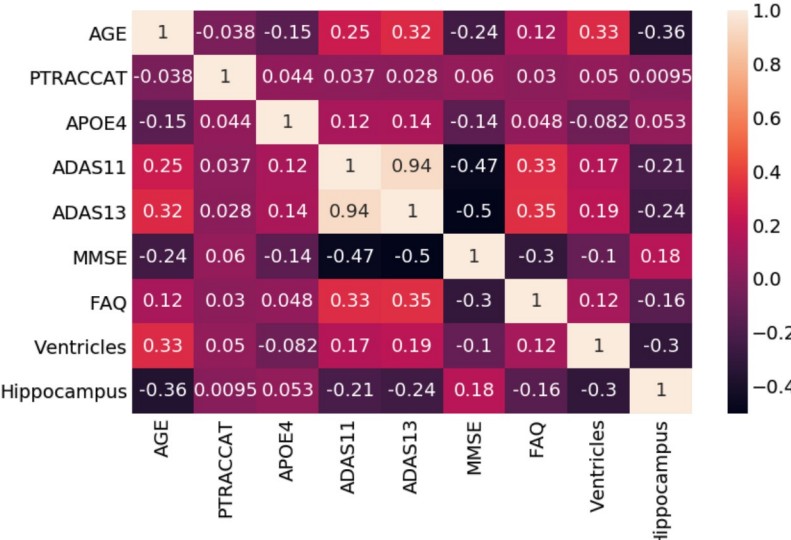

**Fig 4. Random forest correlation matrix.**

MMSE features were found not to be significantly different. The relationships between our features are further seen in Fig 4 as a Correlation Matrix.

## Model performance

A workflow of our random forest model can be seen in Fig 2. This summarizes the training methodology as well as the prediction and variable importance output. After our pre-processing steps, we train a 1000 tree random forest model on 2412 exam visits against different feature groups to compare their results. While initially, using all 13-features seemed to provide the highest accuracy at 91.6%, we found that by removing the RAVLT features, our accuracy rose to 93.6% in the 9-feature group. We also tested a 6-feature group, which removed the RAVLT variables as well as the biomarker data; however, this saw the worst accuracy of the three groups at 89.2%.

After running these feature group variations through our random forest training process, we decided to implement the same training data into a support vector classifier (SVC), an XGBoost classifier, and a Logistic Regression model for comparison (See Table 4). As these are commonly used for this problem, we considered this to be a reasonable comparative measure to the efficacy of our random forest model.

Support vector classifiers attempt to find the separating hyperplane that maximizes the distance of the closest points to the boundary of the class. These are typically effective in high dimensional spaces and have seen a fair amount of usage within the AD conversion prediction domain [14, 8]. In both the 9-feature and 13-feature groups, we found that our random forest model outperformed our SVC implementation (93.6% vs. 90% and 91.6% vs. 90%, respectively). The SVC did show higher accuracy than the 6-feature RF model; however, the AUC was inferior on all SVC variations. The difference in AUC between our best RF variation (96% AUC) and our best SVC variation (54% AUC) is shown in Fig 5. One observation when observing our SVC model is that it struggled to predict the negative class (conversion to AD) and predominantly chose the majority class. This was not the case with our balanced random forest model which was able to appropriately distinguish between both classes.

**Table 4. Performance of random forest vs support vector classifier.**

| Model/Feature | Accuracy | Precision | Recall | F1 Score | AUC | p-value |
|---|---|---|---|---|---|---|
| **Random Forest** | | | | | | |
| 6-Features | 0.892 | 0.907 | 0.980 | 0.942 | 0.88 | 0.91 |
| 9-Features | **0.936** | **0.952** | 0.978 | **0.965** | **0.96** | **0.71** |
| 13-Features | 0.916 | 0.916 | **0.998** | 0.955 | 0.93 | 0.82 |
| **Support Vector** | | | | | | |
| 6-Features | 0.900 | 0.900 | 1 | 0.948 | 0.52 | - |
| 9-Features | 0.900 | 0.900 | 1 | 0.948 | 0.54 | - |
| 13-Features | 0.900 | 0.900 | 1 | 0.948 | 0.55 | - |
| **Logistic Regression** | | | | | | |
| 6-Features | 0.894 | 0.902 | 0.990 | 0.944 | 0.76 | - |
| 9-Features | 0.892 | 0.903 | 0.985 | 0.942 | 0.75 | - |
| 13-Features | 0.896 | 0.904 | 0.990 | 0.945 | 0.75 | - |
| **XGBoost** | | | | | | |
| 6-Features | 0.898 | 0.904 | 0.993 | 0.946 | 0.87 | - |
| 9-Features | 0.920 | 0.930 | 0.985 | 0.957 | 0.89 | - |
| 13-Features | 0.907 | 0.921 | 0.980 | 0.950 | 0.88 | - |

XGBoost is an implementation of gradient boosted decision trees that has seen success in structured data classification. While not being common in the AD conversion prediction space, we wanted to compare how our feature-selection would be handled by its algorithm. XGBoost resulted in the second-best overall method behind our top RF model and showed significantly better performance than the SVC and Logistic Regression implementations. For the 6-feature group, XGBoost outperformed our RF model (89.8% vs. 89.2%). However, while performing better than SVC and Logistic Regression, the XGBoost model still saw less accuracy than the RF model at both the 9 and 13 feature groups, as seen in Table 4. When comparing AUC, one can see how well XGBoost performed (89%) in relation to SVC (54%) and Logistic Regression (75%).

Finally, Logistic Regression was the last method that we leveraged for comparison. Logistic Regression calculates the probability of an event occurrence and can be used when the target variable is categorical. For this model, we trained individual versions of 6, 9, and 13 features but found them to all exhibit less accuracy than our RF model. Additionally, while its AUC (75%) underperformed in contrast to RF and XGBoost, it did significantly better than our best SVC model (54%). Still, this did not result in a model that was close enough to warrant further consideration for our AD conversion problem.

As mentioned previously, our 9-feature random forest implementation with an accuracy of 93.6% and an AUC of 96% against a 383-patient data set represents our best model. While also using the ADNI data set, Grassi et al. [14] could achieve an AUC of 88% with an SVM that made predictions 3 years prior to AD onset. Huang et al. [8] also attained 80% accuracy and 84.6% AUC with an SVM model against the ADNI data set leveraging both clinical and MRI data looking 5 years prior to AD onset. As our approach differs by using Early Mild Cognitive Impairment patients (EMCI) rather than the broader MCI grouping used by other studies, we can predict conversion from 5-7 years prior to the onset of AD. Our outcome is state-of-the-art when comparing our accuracy and AUC to the previously published work for MCI-to-AD prediction as shown in Table 5.

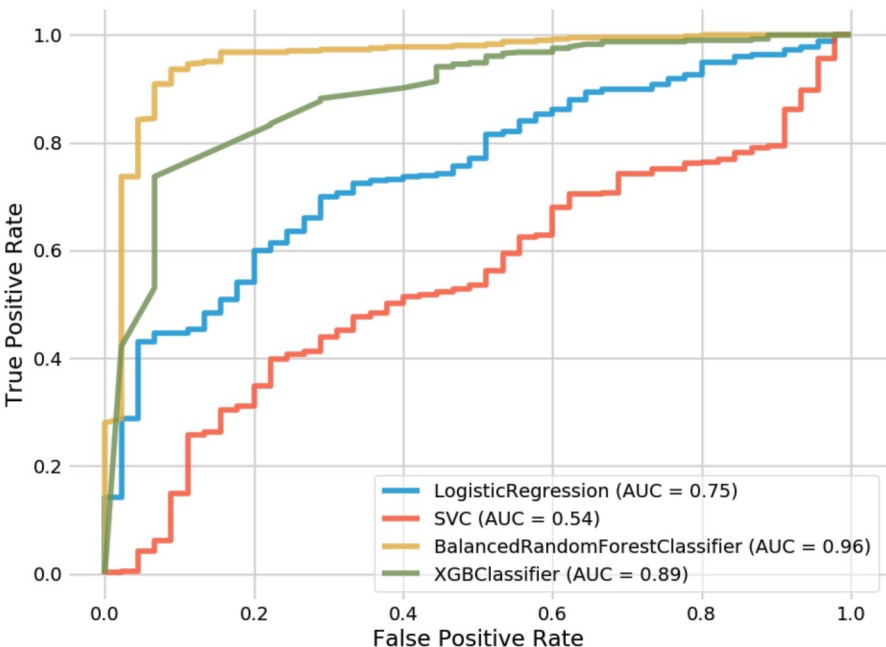

**Fig 5. Receiver operating characteristic curves for random forest and comparison models.**

## Balancing the data

As a result of our imbalanced data set, where 12.8% of the patients belonged to the minority class (EMCI_C); we perform a random oversampling algorithm that generates new samples with replacement from the EMCI_C class. Replacement ensures that samples can be selected, added to the augmented data set, and then returned to the non-augmented data as eligible for further random sampling. For our study, we choose a minority strategy such that all samples would be generated solely from the existing EMCI_C data. This augmentation provides a balance between the two classes so that the majority class does not take over during model training. Through this process of oversampling our number of minority class visits become equivalent to that of the majority. This allows for the model to be trained against 2412 exam visits (1206 per class) rather than only the 1354 from the original data set train/test split. Table 1 further demonstrates the evolution of the data set after oversampling.

**Table 5. State of the art MCI-to-AD prediction.**

| Approach | Data | #Subject | Model | Estimator | MCI-AD(%) | | Predict | Train |
|---|---|---|---|---|---|---|---|---|
| | | | | | ACC | AUC | Year | Time |
| Proposed (ours) | Clinical | ADNI(383) | RF | 1000 | **93.6** | **96** | **5-7** | **2.98 sec** |
| Grassi [14] | Clinical | ADNI(550) | SVM | - | - | 88 | 3 | 2 days |
| Huang [8] | Clinical/MRI | ADNI(290) | SVM | 1000 | 80 | 84.6 | 5 | - |
| Albright [13] | Clinical | ADNI(1737) | MLP | - | - | 86.6 | 5 | - |
| Moore [9] | Clinical | ADNI(1627) | RF | 60 | 73 | 82 | 5 | - |
| Ghazi [15] | MRI | ADNI(742) | RNN | 1000 | - | 76 | 5 | 340 sec |
| Rana [10] | Clinical/MRI | ADNI(559) | CNN | 100 | 69.8 | 83 | 5 | - |
| Thushara [11] | Clinical | ADNI(NA) | RF | 100 | 47.2 | - | 5 | - |

**Table 6. Comparison of imbalanced data set sampling methods.**

| Method | Accuracy |
|---|---|
| Random Over sampler | **93.60** |
| SMOTE | 92.97 |
| Borderline SMOTE | 93.19 |
| ADASYN | 93.30 |

We first compared our random oversampling method against an under-sampling method that targeted the majority class and found a 3.1% increase in accuracy via the oversampling process (see Table 6). Our approach was also compared to class weight modifications, but they performed poorly in comparison to our minority strategy. After determining that oversampling was the preferred method we began to compare against the established oversampling methods.

SMOTE, or rather Synthetic Minority Oversampling Technique, was the first of these methods that we evaluated against. SMOTE relies on generating new information from the minority class population, rather than duplicating from that population. This is done by pulling from a random minority class sample, and then also finding a random k-nearest neighbor from that sample. The new data is then created in a space between those two samples [16]. However, against our EMCI data set this was found to reduce accuracy by .63% compared to our original technique.

Borderline SMOTE was also considered as this modifies the SMOTE technique to generate new data along the decision boundary of the two classes, rather than randomly between two samples [17]. While we did see a 0.22% improvement over SMOTE, it still fell short of our Random Oversampler.

Finally, we attempted the Adaptive Synthetic Sampling (ADASYN) method as a means of comparison. This deviates from the other SMOTE methods by generating new data based on the density of the data, rather than the decision boundary or k-nearest neighbor. ADASYN focuses its synthetic data creation within the low density feature space regions and creates less data within the high density regions [18]. For our data, this method produced the second best results and outperformed both SMOTE and Borderline SMOTE. The overall accuracy comparison of these oversampling techniques can be seen in Table 6.

## Assessment of model feature importance

One advantage of using the random forest algorithm is that feature importance can be assessed at both the model and individual prediction levels. The model feature importance of our three feature groupings can be seen in Figs 6–8. As a random forest algorithm deals with different combinations of features in each of its' decision trees, this allows for the feature importance to be calculated based on how much the prediction error increases [9]. This is done by first calculating the individual nodes' importance per tree as seen in Eq 1. Within this, $ni_j$ represents the importance of node $j$, $w_j$ being the weighted samples reaching node j, and $C_j$ as the impurity value of the node. Once each node's importance has been determined, the feature importance per tree is calculated per Eq 2 and is then normalized to a value between 0 and 1 per Eq 3. This result is then averaged across the entire forest before being divided by the total number of trees

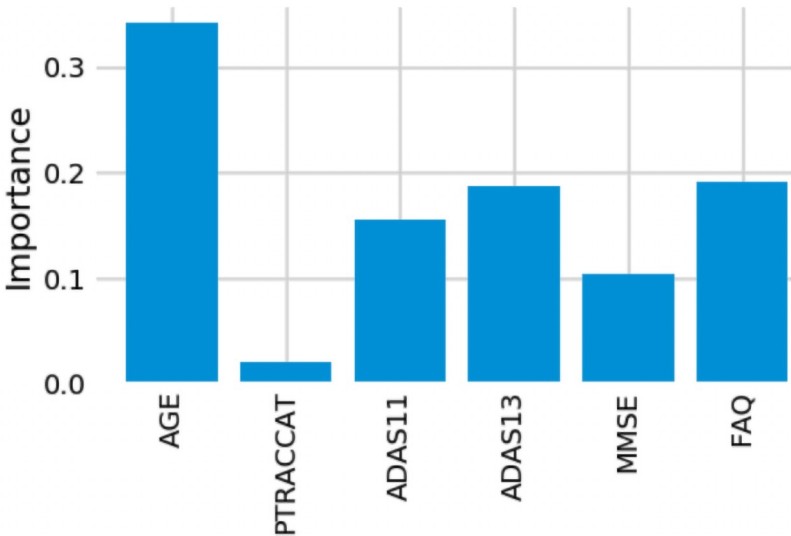

**Fig 6. 6-Feature model feature importance.**

[19].

$$ni_j = w_j C_j - w_{left(j)} C_{left(j)} - w_{right(j)} C_{right(j)} \tag{1}$$

$$fi_i = \frac{\sum_{j:node\ j\ splits\ on\ feature\ i} ni_j}{\sum_{k \in\ all\ nodes} ni_k} \tag{2}$$

$$normfi_i = \frac{fi_i}{\sum_{j \in\ all\ features} fi_j} \tag{3}$$

$$RFfi_i = \frac{\sum_{j \in\ all\ trees} norm\ fi_{ij}}{T} \tag{4}$$

For our 6-feature model, the three most important features are Age, FAQ, and ADAS13. For our top-performing 9-feature model, the top features are Age, Hippocampus, and Ventricles. Finally, for our 13-feature model, Age, Hippocampus, and FAQ score are the most important. The presence of hippocampal and ventricular volume towards the top explains why the absence of those features in our 6-feature model resulted in diminished accuracy. By adding in the RAVLT features, our accuracy improved, but these were redundant with the other neuropsychological scale scores, so they were removed for our final model. Age was consistently seen as the best conversion predictor, which corresponds to the increased risk of AD at an older age [20]. Race (PTRACCAT) was routinely at the lowest feature importance between our models, but we did observe a decrease in accuracy upon its' removal. This is likely due to race having very little correlation with the other included features, whereas some neuropsychological scores exhibited signs of possible overlap (RAVLT).

For our best model, we also assess the permutation importance seen in Eq 3. This reduces the high cardinality bias seen in the feature importance charts by permuting against a held-out test set. This is done by each feature column being permuted against a baseline metric that was

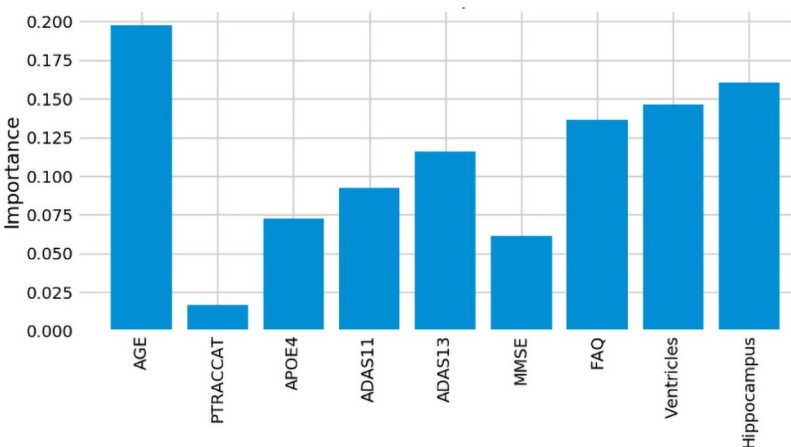

**Fig 7. 9-Feature model feature importance.**

initially evaluated against the data set. The permutation importance is then established as the difference between the baseline metric and the feature column permutation. From this, we see that Age and FAQ maintain their high importance. However, APOE4 is now significantly more relevant in regards to the test set prediction (see Fig 9).

Table 7 shows the average standard deviation differences from the mean out of the subjects that the model predicted incorrectly. We omit our PTRACCAT (Race) feature into this analysis, given that it is not a continuous variable. In total, 20 EMCI_C ground truth subjects and 9 EMCI_NC ground truth subjects were incorrectly classified. By analyzing the standard deviation differences, we can determine which feature was most abnormal compared to the average model prediction for that given class.

We do this by establishing the data set means and standard deviations per feature for both the EMCI_C and EMCI_NC classes. We then take each misclassified patient's feature values and subtract them by the corresponding mean, prior to dividing them by that feature's standard deviation value. This allows us to see which features were the most unusual at an individual patient level. Coupled with the feature importance ranking this gives us clearer insight into the model's prediction rationale. For example, the MMSE feature was 2.37 standard deviations away from its EMCI_C mean, which contributed to our model misclassifying those cases as

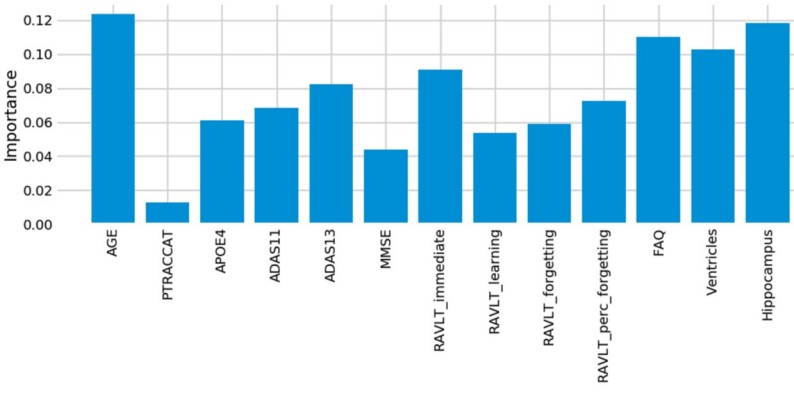

**Fig 8. 13-Feature model feature importance.**

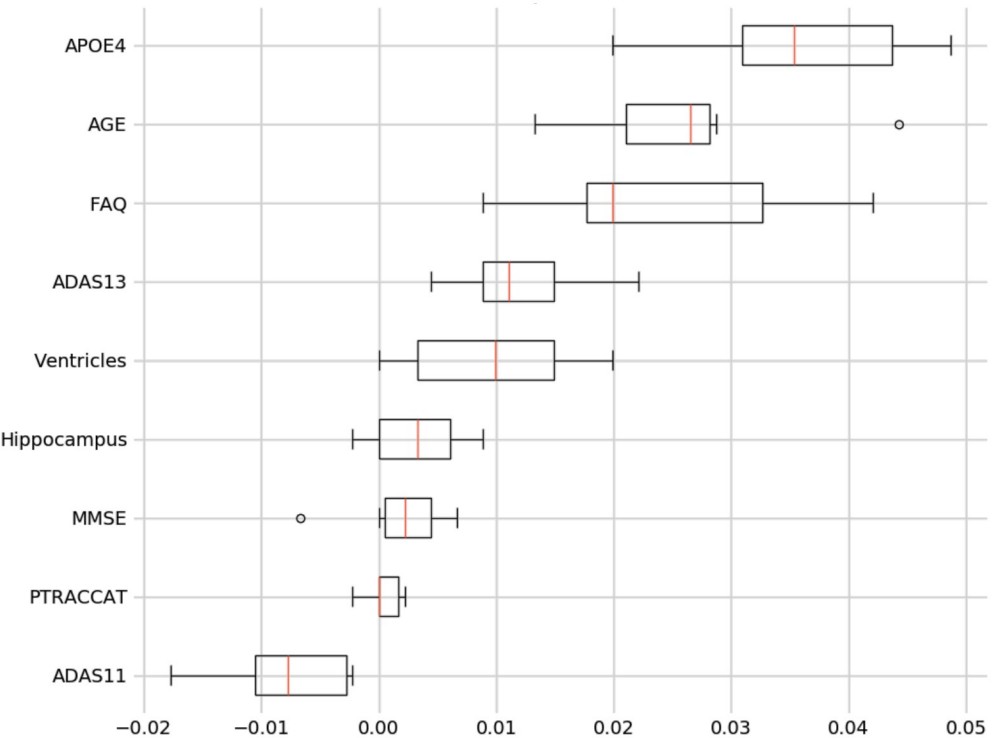

**Fig 9. 9-Feature model permutation importance.**

EMCI_NC. Of the EMCI_C misclassifications, MMSE proved to be the most misleading feature. However, we still saw higher overall accuracy by including it within our model because of its relatively low feature importance.

Out of the EMCI_NC misclassifications, there was less clarity as to which feature was problematic. However, we do observe a standard deviation increase in our neuropsychological test scores across the nine misclassifications. This is indicative of the model believing these subjects' test scores to be similar to those of the EMCI_C class and thus making the false prediction. In future work, we will explore whether knowing these misleading instances can help our model's accuracy, but currently, it appears that these are outliers within the ADNI dataset.

## Assessment of individual predictors' feature importance

For the individual level, we can see which features specific to that patient made the largest difference contribution to their prediction. An example of these prediction contributions can be seen in Fig 10 based on the test patient's features provided in Tables 8 and 9. In this case, our model correctly predicted that this patient would convert to AD with an overall confidence of 90.4%. This confidence is a reflection of the aggregate of all of the individual trees' votes within our forest.

**Table 7. Average standard deviation difference for incorrect predictions by ground truth class and feature.**

|         | Record# | Age  | APOE4 | ADAS11 | ADAS13 | MMSE | FAQ  | Ventricles | Hippocampus |
|---------|---------|------|-------|--------|--------|------|------|------------|-------------|
| **EMCI_C**  | 20      | 1.90 | 2.20  | 1.30   | 1.38   | **2.37** | 0.82 | 0.80       | 1.00        |
| **EMCI_NC** | 9       | 1.74 | 2.26  | **2.87** | 2.46   | 2.68 | 2.61 | 1.13       | 2.76        |

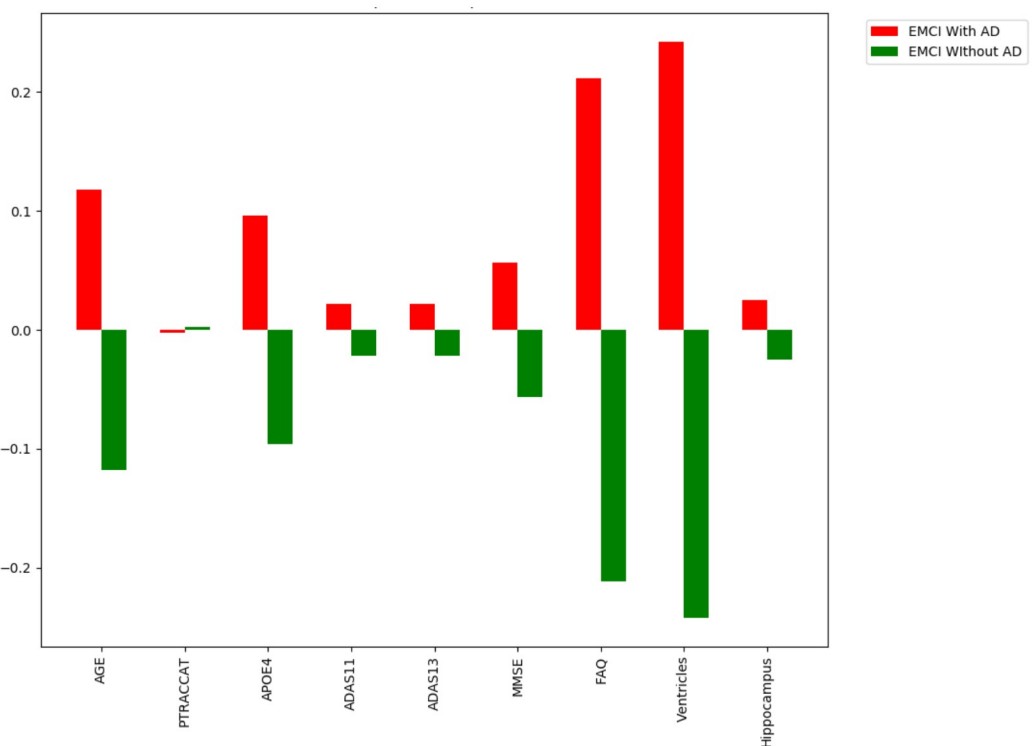

**Fig 10. Feature importance example for an individual patient.**

From Fig 10 and the contributions listed in Tables 8 and 9, we see that ventricular volume was the most essential predictive feature for this patient as it contributed 24.2% towards the model's decision prediction. This was closely followed by their Functional Activities Questionnaire (FAQ) score which contributed 21.2% of the prediction. Race (PTRACCAT) was the only feature that contributed to the wrong prediction, albeit only a 0.3% prediction contribution for this patient. These individual feature importances are calculated in an identical manner to the model feature importance formulas above, however they do not include the model-

**Table 8. Example features and prediction contributions (9FT PC) for EMCI_C cases.**

| EMCI_C Class | | Correctly Classified | | | | Misclassified | | | |
|---|---|---|---|---|---|---|---|---|---|
| 9 Features | Mean | Subject C1 | | Subject C2 | | Subject C3 | | Subject C4 | |
| | | Value | PC | Value | PC | Value | PC | Value | PC |
| **Age** | 73.5 | 68 | 0.11 | 77 | **0.118** | 69.1 | 0.008 | 73 | **0.337** |
| **FAQ** | 4.1 | 0 | -0.036 | 10 | **0.212** | 4 | 0.007 | 3 | 0.039 |
| **ADAS13** | 15.8 | 20 | 0.042 | 17 | 0.022 | 10 | -0.008 | 9 | 0.025 |
| **ADAS11** | 9.7 | 13 | 0.039 | 9 | 0.022 | 5 | -0.006 | 4 | 0.013 |
| **MMSE** | 28.1 | 29 | 0.006 | 26 | 0.057 | 26 | 0.023 | 29 | -0.011 |
| **PTRACCAT** | – | 7 | -0.001 | 7 | -0.003 | 7 | 0.003 | 7 | 0 |
| **Hippocampus** | 6875.2 | 7853 | **0.173** | 6901 | 0.025 | 5576 | **0.158** | 7835 | -0.02 |
| **Ventricles** | 39282.7 | 38627 | **0.145** | 24285 | **0.242** | 35280.12 | -0.006 | 32379 | -0.031 |
| **APOE4** | 0.9 | 2 | **0.248** | 1 | 0.096 | 2 | **0.138** | 0 | -0.022 |
| **PC: Sum (AVG)** | | **0.726 (0.08)** | | **0.79 (0.087)** | | **0.317 (0.035)** | | **0.33 (0.036)** | |

**Table 9. Example features and prediction contributions (9FT PC) for EMCI_NC cases.**

| EMCI_NC Class | | Correctly Classified | | | | Misclassified | | | |
|---|---|---|---|---|---|---|---|---|---|
| 9 Features | Mean | Subject NC1 | | Subject NC2 | | Subject NC3 | | Subject NC4 | |
| | | Value | PC | Value | PC | Value | PC | Value | PC |
| Age | 71.1 | 59 | 0.047 | 81 | 0.001 | 79.8 | **0.106** | 63.6 | 0.009 |
| FAQ | 1.82 | 4 | -0.037 | 1 | 0.056 | 5 | 0.099 | 17 | **0.158** |
| ADAS13 | 13.3 | 16 | -0.001 | 21 | -0.005 | 17 | -0.013 | 37 | **0.346** |
| ADAS11 | 8.5 | 13 | -0.002 | 14 | 0.009 | 12 | 0.019 | 27 | **0.208** |
| MMSE | 28.3 | 29 | 0.015 | 27 | -0.008 | 30 | -0.002 | 19 | -0.025 |
| PTRACCAT | – | 7 | 0.001 | 7 | 0.008 | 7 | -0.006 | 7 | -0.002 |
| Hippocampus | 7334.1 | 8303 | 0.023 | 6288 | 0.015 | 5437 | **0.256** | 7223.86 | -0.021 |
| Ventricles | 34504.6 | 22275 | -0.03 | 30260 | -0.001 | 69583 | -0.043 | 35280.12 | -0.019 |
| APOE4 | 0.4 | 0 | 0.045 | 1 | -0.002 | 0 | -0.041 | 0 | -0.037 |
| PC: Sum (AVG) | | 0.061 (0.006) | | 0.073 (0.008) | | 0.375 (0.041) | | 0.617 (0.068) | |

level aggregation. In this sense, it allows us to interpret precisely why the decision trees chose a certain classification.

Tables 8 and 9 show the prediction contributions from subjects across both the EMCI_C and EMCI_NC classes. The PC column represents the amount of each feature's contribution to the overall prediction. A positive value indicates the contribution towards the ground truth class, whereas a negative value denotes the contribution towards the incorrect class. The PC's overall sum and average for the EMCI_C class's correctly classified cases are higher than 0.7 and 0.08 while ones for the EMCI_C class are lower than 0.061 and 0.008, respectively. The PC sum and average are strong indicators for the classification. As Table 7 is helpful for interpreting possibly misleading features at the model level, this individual PC metric allows us to better understand the model's decision making on a per subject basis.

## Discussion

We have demonstrated that a random forest model can take clinical features and accurately predict MCI-to-AD conversion probability. Our RF classifier showed superior performance compared to competing SVM, XGBoost, and Logistic Regression implementations, including our own. It is also worth noting that the best models looked at all MCI patients, rather than the earlier EMCI subset. This gives our model the strength of predicting from 5-7 years prior to the onset of AD. Our results show that clinical features can also outperform MRI-based models. This is important as obtaining neuropsychological scores, a significant subset of our chosen features can be far more affordable and less intensive than obtaining a patient's MRI imaging. With a more flexible approach, the expectation is that this predictor would be easier to deploy into a clinical setting.

In our experiments with the feature groupings, we found neuropsychological scores to be the most reliable and essential feature subset as we always experienced lower model accuracy with their exclusion. Performing tests on individual predictors also showed their weaknesses as each predictor demonstrated improved accuracy when coupled with an additional predictor. Even the neuropsychological scores by themselves exhibited signs of subjectivity, which were remediated by including biomarker and demographic features.

Additionally, our methods for oversampling an initially imbalanced data set can be of use throughout the medical research domain. With many medical data sets consisting of similar target class imbalance, our process enhances bagging algorithms by augmenting more samples

for the minority classes. For our purpose, this was only tested within a binary classification problem, however we will be extending this technique to multi-class problems.

One limitation of this study is that all of the patients were from the ADNI data set. While our accuracy was verified by splitting our data across multiple instances, we did not test the population outside of the ADNI participants. The inclusion of other data sets into our model would help account for even more significant variations and will be a target for future work.

In the future, we would like to combine this clinical features dependent model with our prior diffusion tensor imaging model [21] in order to create an ensemble predictor that can handle a large variety of available patient information. This would allow for greater flexibility for patient input data while maintaining high accuracy in the prediction. Additionally, it is currently difficult to differentiate between the sub-types of dementia when a patient presents with cognitive and memory decline [22]. This can lead to an inaccurate treatment plan if the patient is misdiagnosed. Having the ability to predict additional sub-types at such an early stage would help significantly with pharmacological management [23]. Researching the differences between these sub-types based on this study's clinical features will be a subject of our future work.

In summary, we created a balanced random forest model based on multiple features to predict the MCI-to-AD conversion probability. In addition, we determined which features were most important for the overall model, as well as for individual patient predictions. We also took advantage of oversampling methods to better balance the target classes. As early detection is critical for both clinical trial enrollment and cost-effective treatment plans, we expect our work to help in clinical diagnosis as well as establishing treatment timelines. Our random forest model achieved state-of-the-art performance with an accuracy of 93.6% and showed that the combination of demographic, neuropsychological scores and biomarker features could be used to predict which EMCI patients are at a higher risk of AD.

## Acknowledgments

Data used in preparation of this manuscript were obtained from the Alzheimer's Disease Neuroimaging Initiative (ADNI) database (adni.loni.usc.edu). The ADNI investigators contributed to the design of the ADNI source data but was not involved in any other aspect of this work. A complete listing of ADNI investigators can be found at: http://adni.loni.usc.edu/study-design/ongoing-investigations/.

## Author Contributions

**Conceptualization:** Matthew Velazquez, Yugyung Lee.

**Data curation:** Matthew Velazquez.

**Formal analysis:** Matthew Velazquez.

**Investigation:** Matthew Velazquez.

**Methodology:** Matthew Velazquez, Yugyung Lee.

**Project administration:** Matthew Velazquez.

**Resources:** Matthew Velazquez.

**Software:** Matthew Velazquez.

**Supervision:** Yugyung Lee.

**Validation:** Matthew Velazquez.

**Visualization:** Matthew Velazquez.

**Writing – original draft:** Matthew Velazquez.

**Writing – review & editing:** Yugyung Lee.

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
