## [Decision Letter · Decision Letter 0]

14 Jan 2021

PONE-D-20-39339

Random forest model for feature-based Alzheimer's disease conversion prediction from early mild cognitive impairment subjects

PLOS ONE

Dear Dr. Velazquez,

Thank you for submitting your manuscript to PLOS ONE. After careful consideration by 2 Reviewers and an Academic Editor, all of the critiques of both Reviewers must be addressed in detail in a revision to determine publication status. If you are prepared to undertake the work required, I would be pleased to reconsider my decision, but revision of the original submission without directly addressing the critiques of the 2 Reviewers does not guarantee acceptance for publication in PLOS ONE. If the authors do not feel that the queries can be addressed, please consider submitting to another publication medium. A revised submission will be sent out for re-review. The authors are urged to have the manuscript given a hard copyedit for syntax and grammar.

**Comments to the Author**

1. Is the manuscript technically sound, and do the data support the conclusions?

Reviewer #1: Partly

Reviewer #2: Partly

2. Has the statistical analysis been performed appropriately and rigorously? 

Reviewer #1: Yes

Reviewer #2: Yes

3. Have the authors made all data underlying the findings in their manuscript fully available?

Reviewer #1: Yes

Reviewer #2: Yes

4. Is the manuscript presented in an intelligible fashion and written in standard English?

Reviewer #1: Yes

Reviewer #2: Yes

5. Review Comments to the Author

Reviewer #1: The authors use random forests - a common technique in machine learning - to classify EMCI cases from the ADNI database in terms of whether or not they will progress to AD, using features that are available in the database (e.g., age, hippocampal volume, MMSE...). The approach produces good results (>93% accuracy) that improve upon other results in the literature. Overall, the paper is interesting and makes a worthy contribution to the field of machine learning applied to Alzheimer Disease. Please see my comments below.

(1) There is an inconsistency (repeated three times in the paper) in terms of how many cases were used. On the one hand, the authors say they used 383 EMCI cases. However, they break it down into 49 cases that converted to AD, and 335 that did not convert, which doesn't add up.

(2) The data set was broken up into 288 training and 95 test cases (again, this is not equal to 49+335). However, the authors did not specify how the split was done (presumably at random?). Was any cross-validation used?

(3) Since there is a statistically significant difference in age between the patients in the database who converted to AD and those who didn't, did the authors consider creating a smaller dataset where age would be matched between the two groups?

(4) The approach of balancing data by oversampling the patients who converted to AD was interesting.

(5) The discussion on feature importance was nicely presented. However, to improve clarity, I suggest that the way of calculating importance (p. 8, lines 202-204) be presented in mathematical form rather than in words. Is this a common way of reporting feature importance for RF algorithms?

(6) In Table 2, it might be insightful to report ranges for the number of years between first diagnosis and conversion to AD (for those who converted), and number of years of follow-up (for those who didn't).

(7) Overall, the paper is well written in clear and concise language. There are just a few typos:

p. 2, line 16: "Random Forest is..." -> "A Random Forest is..."

p. 2, line 18: "...and has been..." -> "...and that has been..."

p. 2, line 18: "In our work, this Random..." -> "In our work, a Random..."

p. 2, line 22: "Through Random Forest..." -> "Through a Random Forest..."

(8) I notice that the references appear to be from 2019 and earlier. Given the incredible number of publications coming out in machine learning applied to clinical problems, I strongly urge the authors to search once more for recent (2019 and 2020) papers, say on using random forests in dementia (including those that make use of imaging such as MRI or PET). This will make the Reference list more complete than it currently is.

(9) For completeness, I encourage the authors to explicitly state the number of trees in their RF in the abstract (and possibly in Table 2, even if it means leaving blank entries for the other approaches from the literature), and to report some measure of computational cost (say, run time to train the RF).

Reviewer #2: The paper aims at devising a method that will correctly predict probability of MCI to AD conversion.

1. The dataset for EMCI_C is extremely small compared to the non converted class. Although, the authors provide a brief explanation of circumventing the imbalanced dataset issue, it is not enough. Precisely, how was the oversampling and undersampling done? Depending on the sampling technique, the prediction performance can vary. Did the authors try to use existing methods, for example, SMOTE, SMOTEBoost, SHRINK or compare theirs against these? This section deserves more attention because the conclusions drawn in the paper would not have been possible with the imbalanced nature of the raw dataset. Thus proper methods and relevant explanations are required. Also, justification needs to be provided as to why the number of visits were optimized to do the oversampling and undersampling?

2. Using this classifier, how many months prior to conversion can the MCI to AD conversion be predicted?

3. The authors have used a logistic regression model as one of the competitors for RandomForest. Please use the 6 feature selection, 9 feature selection, and other variables as covariates in the Logistic regression model and then estimate the accuracy of conversion. That way, the benefits of a logistic regression model van be fully exploited and a fair comparison between randomForests and Logistic regression would have been done.

4. The individual wise prediction is a little too far fetched with too many variables being accounted for in a single individual and contribution of different factors estimated for the prediction. Also this is just a set of 5 individuals out of an already small dataset. So better to remove this segment from the paper.

6. PLOS authors have the option to publish the peer review history of their article (what does this mean?). If published, this will include your full peer review and any attached files.

**Do you want your identity to be public for this peer review?** For information about this choice, including consent withdrawal, please see our Privacy Policy.

Reviewer #1: No

Reviewer #2: No

We look forward to receiving your revised manuscript.

Kind regards,

Stephen D. Ginsberg, Ph.D.

Section Editor

PLOS ONE

"Data collection and sharing for this project was funded by the Alzheimer's Disease

Neuroimaging Initiative (ADNI) (National Institutes of Health Grant U01 AG024904) and DOD ADNI (Department of Defense award number W81XWH-12-2-0012). ADNI is

funded by the National Institute on Aging, the National Institute of Biomedical

Imaging and Bioengineering, and through generous contributions from the following:

AbbVie, Alzheimer's Association; Alzheimer's Drug Discovery Foundation; Araclon

Biotech; BioClinica, Inc.; Biogen; Bristol-Myers Squibb Company; CereSpir, Inc.;

Cogstate; Eisai Inc.; Elan Pharmaceuticals, Inc.; Eli Lilly and Company; Eurolmmun; F.

Hoffmann-La Roche Ltd and its affiliated company Genentech, Inc.; Fujirebio; GE

Healthcare; IXICO Ltd.; Janssen Alzheimer Immunotherapy Research & Development

LLC.; Lumosity; Lundbeck; Merck & Co., Inc.; Meso Scale Diagnostics, LLC.; NeuroRx

Research; Neurotrack Technologies; Novartis Pharmaceuticals Corporation; Pfizer Inc.;

Piramal Imaging; Servier; Takeda Pharmaceutical Company; and Transition

Therapeutics. The Canadian Institutes of Health Research is providing funds to support

ADNI clinical sites in Canada. Private sector contributions are facilitated by the

Foundation for the National Institutes of Health [18]. The grantee organization is the

Northern California Institute for Research and Education, and the study is coordinated

by the Alzheimer's Therapeutic Research Institute at the University of Southern

California."

Additionally, because some of your funding information pertains to commercial funding, we ask you to provide an updated Competing Interests statement, declaring all sources of commercial funding.

In your Competing Interests statement, please confirm that your commercial funding does not alter your adherence to PLOS ONE Editorial policies and criteria by including the following statement: "This does not alter our adherence to PLOS ONE policies on sharing data and materials.” as detailed online in our guide for authors  http://journals.plos.org/plosone/s/competing-interests.  If this statement is not true and your adherence to PLOS policies on sharing data and materials is altered, please explain how.

Please include the updated Competing Interests Statement and Funding Statement in your cover letter. We will change the online submission form on your behalf.

4. One of the noted authors is a group or consortium [Alzheimer's Disease Neuroimaging Initiative™]. In addition to naming the author group, please list the individual authors and affiliations within this group in the acknowledgments section of your manuscript. Please also indicate clearly a lead author for this group along with a contact email address.

---

## [Author Response · Author response to Decision Letter 0]

25 Feb 2021

Dear Editor, 

We are pleased to submit a revised version of our “Random Forest Model for Feature-based Alzheimer's Disease Conversion Prediction from Early Mild Cognitive Impairment Subjects” manuscript for consideration at PLOS ONE. The comments provided by the reviewers were very helpful in strengthening the structure and idea of the paper. 

We have individually addressed the feedback received and have made considerable changes to the content of the work. Please find our responses in blue below for each reviewer comment. 

Both authors have been involved in these revisions and have approved the resubmission. We hope that these revisions address the reviewers’ concerns, and that this version is more fitting for publication. Thank you very much for taking the time to review our work in-depth and we look forward to your response.

Sincerely,

Matthew Velazquez & Dr. Yugyung Lee

 

Reviewer 1: 

(1) There is an inconsistency (repeated three times in the paper) in terms of how many cases were used. On the one hand, the authors say they used 383 EMCI cases. However, they break it down into 49 cases that converted to AD, and 335 that did not convert, which doesn't add up.

Thanks for catching this. We have corrected the typos to 334 for the EMCI_NC amount.

(2) The data set was broken up into 288 training and 95 test cases (again, this is not equal to 49+335). However, the authors did not specify how the split was done (presumably at random?). Was any cross-validation used?

Thanks again for finding this inconsistency. As above, we have corrected the typos to reflect the correct sum. We have also added text to detail the randomness of the training/validation split within our methods section as this was previously missing. Regarding cross-validation, we had originally included it in our code as a training output along with the confusion matrix and AUC. However, upon further literature review it appears that cross-validation is not necessary with Random Forest due to the multiple bagging approach that it uses. As a result, we chose to leave it out of the main body but could include that if desired. Here are a few of the discussions that we had observed:

https://www.kaggle.com/c/titanic/discussion/10089

https://stats.stackexchange.com/questions/283760/is-cross-validation-unnecessary-for-random-forest

https://stats.stackexchange.com/questions/365201/k-fold-on-a-random-forest

(3) Since there is a statistically significant difference in age between the patients in the database who converted to AD and those who didn't, did the authors consider creating a smaller dataset where age would be matched between the two groups?

Thank you for this suggestion as we had not considered this initially. To account for this we trained on a smaller dataset that eliminated Age outliers until the Age class difference was not statistically significant. Additionally, the Age difference was not statistically significant when measuring against Feature Importance between classes. We have modified the Demographic and Clinical Characteristics section to include these new observations. 

(4) The approach of balancing data by oversampling the patients who converted to AD was interesting.

Thank you! We have added significantly more detail regarding our over-sampling algorithm method as well as a comparison against others within the ‘Balancing the Data’ sub-section.

(5) The discussion on feature importance was nicely presented. However, to improve clarity, I suggest that the way of calculating importance (p. 8, lines 202-204) be presented in mathematical form rather than in words. Is this a common way of reporting feature importance for RF algorithms?

Per your suggestion, we have added formulas detailing the feature importance calculations to the ‘Assessment of model feature importance’ sub-section as well as expanding the text to improve clarity as to how the importance is derived.

(6) In Table 2, it might be insightful to report ranges for the number of years between first diagnosis and conversion to AD (for those who converted), and number of years of follow-up (for those who didn't).

Thank you for this suggestion as this will help the readers understand the data set better. Per your suggestion, we have added Time to AD Conversion and Time in Study rows to Table 2.

(7) Overall, the paper is well written in clear and concise language. There are just a few typos:

p. 2, line 16: "Random Forest is..." -> "A Random Forest is..."

p. 2, line 18: "...and has been..." -> "...and that has been..."

p. 2, line 18: "In our work, this Random..." -> "In our work, a Random..."

p. 2, line 22: "Through Random Forest..." -> "Through a Random Forest..."

We have revised this verbiage per your recommendations.

(8) I notice that the references appear to be from 2019 and earlier. Given the incredible number of publications coming out in machine learning applied to clinical problems, I strongly urge the authors to search once more for recent (2019 and 2020) papers, say on using random forests in dementia (including those that make use of imaging such as MRI or PET). This will make the Reference list more complete than it currently is.

We appreciate this comment as it allowed for us to expand our Related Work section to present a stronger argument of our model’s differences. Upon further literature review, we added the following references to our work:

10.Rana SS, Ma X, Pang W, Wolverson E. A Multi-Modal Deep Learning Approachto the Early Prediction of Mild Cognitive Impairment Conversion to Alzheimer’sDisease. Institute of Electrical and Electronics Engineers (IEEE); 2020. p. 9–18.

11. Thushara A, Ushadevi Amma C, John A, Saju R. Multimodal MRI BasedClassification and Prediction of Alzheimer’s Disease Using Random ForestEnsemble. In: Proceedings - 2020 Advanced Computing and CommunicationTechnologies for High Performance Applications, ACCTHPA 2020. Institute ofElectrical and Electronics Engineers Inc.; 2020. p. 249–256.

Additionally, we have included them as part of our results comparison in Table 5.

(9) For completeness, I encourage the authors to explicitly state the number of trees in their RF in the abstract (and possibly in Table 2, even if it means leaving blank entries for the other approaches from the literature), and to report some measure of computational cost (say, run time to train the RF).

Per your suggestion, we have added the number of estimators to our abstract section. We have also added these estimators to Table 5 alongside a new Training Time column that reflects the computational cost. Thanks again for all the feedback!

Reviewer 2: 

1. The dataset for EMCI_C is extremely small compared to the non converted class. Although, the authors provide a brief explanation of circumventing the imbalanced dataset issue, it is not enough. Precisely, how was the oversampling and undersampling done? Depending on the sampling technique, the prediction performance can vary. Did the authors try to use existing methods, for example, SMOTE, SMOTEBoost, SHRINK or compare theirs against these? This section deserves more attention because the conclusions drawn in the paper would not have been possible with the imbalanced nature of the raw dataset. Thus proper methods and relevant explanations are required. Also, justification needs to be provided as to why the number of visits were optimized to do the oversampling and undersampling?

Thanks for the comments as this helped us bolster our methods section. We have expanded greatly upon the Balancing the Data section and have included comparisons to SMOTE, Borderline SMOTE, and ADASYN. This involved retraining the model with each of those methods and then reporting their results in Table 3. We have also included additional text to improve clarity regarding the pre-augmentation and post-augmentation visits as well as more detail on the Random Oversampler algorithm within the same section. 

2. Using this classifier, how many months prior to conversion can the MCI to AD conversion be predicted?

Since our classifier is trained on Early MCI patients rather than the broader MCI, we are able to predict conversion from 5-7 years prior to AD diagnosis. We have added this range as a new column within our Table 5 comparisons.

3. The authors have used a logistic regression model as one of the competitors for RandomForest. Please use the 6 feature selection, 9 feature selection, and other variables as covariates in the Logistic regression model and then estimate the accuracy of conversion. That way, the benefits of a logistic regression model van be fully exploited and a fair comparison between randomForests and Logistic regression would have been done.

We have added additional text within the Methods section to clarify the feature grouping training for the Logistic Regression models that we evaluated. In addition, we compare these versions in Table 4.

4. The individual wise prediction is a little too far fetched with too many variables being accounted for in a single individual and contribution of different factors estimated for the prediction. Also this is just a set of 5 individuals out of an already small dataset. So better to remove this segment from the paper.

We appreciate that you brought this to our attention. As a result, we have drastically reduced the speculative text that was previously present within the “Assessment of individual predictors’ feature importance” section. We have also redesigned our existing tables into two new tables (8 and 9), which displays patient examples stratified by class. In combination with the new Feature Importance formulas and the ranking table (Table 6), this allows for a more mathematical and less speculative assessment to be inferred by the reader.

---

## [Decision Letter · Decision Letter 1]

9 Mar 2021

PONE-D-20-39339R1

Random forest model for feature-based Alzheimer's disease conversion prediction from early mild cognitive impairment subjects

PLOS ONE

Dear Dr. Velazquez,

Thank you for resubmitting your work to PLOS ONE. Please make the corrections posed by Reviewer #2 so I can render a decision on this manuscript.

**Comments to the Author**

1. If the authors have adequately addressed your comments raised in a previous round of review and you feel that this manuscript is now acceptable for publication, you may indicate that here to bypass the “Comments to the Author” section, enter your conflict of interest statement in the “Confidential to Editor” section, and submit your "Accept" recommendation.

Reviewer #2: All comments have been addressed

2. Is the manuscript technically sound, and do the data support the conclusions?

Reviewer #2: Yes

3. Has the statistical analysis been performed appropriately and rigorously? 

Reviewer #2: Yes

4. Have the authors made all data underlying the findings in their manuscript fully available?

Reviewer #2: Yes

5. Is the manuscript presented in an intelligible fashion and written in standard English?

Reviewer #2: Yes

6. Review Comments to the Author

Reviewer #2: The authors have addressed points raised during revision.

One thing, the individualized prediction suffers from low sample size, and it remains to be seen how the individualized prediction behaves in other datasets. Old age is the most significant factor for MCI to AD conversion. Therefore, authors might want to keep out the "individualized" viewpoint from the abstract section and consider their handling of this subject matter in the results and discussion.

7. PLOS authors have the option to publish the peer review history of their article (what does this mean?). If published, this will include your full peer review and any attached files.

**Do you want your identity to be public for this peer review?** For information about this choice, including consent withdrawal, please see our Privacy Policy.

Reviewer #2: No

We look forward to receiving your revised manuscript.

Kind regards,

Stephen D. Ginsberg, Ph.D.

Section Editor

PLOS ONE

---

## [Author Response · Author response to Decision Letter 1]

9 Mar 2021

As per Reviewer #2's feedback, I have revised the abstract and also the discussion sections. For the discussion section I have removed some conclusions that were drawn based on the individual case analysis. These will be revisited in future work across additional datasets. Thanks again for the feedback!

---

## [Editor Report · Decision Letter 2]

15 Mar 2021

Random forest model for feature-based Alzheimer's disease conversion prediction from early mild cognitive impairment subjects

PONE-D-20-39339R2

Dear Dr. Velazquez,

We’re pleased to inform you that your manuscript has been judged scientifically suitable for publication and will be formally accepted for publication once it meets all outstanding technical requirements.

Kind regards,

Stephen D. Ginsberg, Ph.D.

Section Editor

PLOS ONE

---

## [Editor Report · Acceptance letter]

13 Apr 2021

PONE-D-20-39339R2 

Random forest model for feature-based Alzheimer’s disease conversion prediction from early mild cognitive impairment subjects 

Dear Dr. Velazquez:

I'm pleased to inform you that your manuscript has been deemed suitable for publication in PLOS ONE. Congratulations! Your manuscript is now with our production department. 

Kind regards, 

on behalf of

Dr. Stephen D. Ginsberg 

Section Editor

PLOS ONE